Revisiting food-sourced vitamins for consumer diet and health needs: a perspective review, from vitamin classification, metabolic functions, absorption, utilization, to balancing nutritional requirements

http://orcid.org/0000-0002-0835-5872 Ofoedu Chigozie E. 1 2 chigozie.ofoedu@futo.edu.ng
Iwouno Jude O. 2
Ofoedu Ebelechukwu O. 2
Ogueke Chika C. 2
Igwe Victory S. 1 2
Agunwah Ijeoma M. 2
Ofoedum Arinze F. 2
Chacha James S. 1 3
Muobike Onyinye P. 2
Agunbiade Adedoyin O. 1 4
Njoku Njideka E. 2
Nwakaudu Angela A. 2
Odimegwu Nkiru E. 2
Ndukauba Onyekachi E. 2
Ogbonna Chukwuka U. 5
http://orcid.org/0000-0001-9273-0366 Naibaho Joncer 6 joncer.naibaho@upwr.edu.pl
Korus Maciej 6
http://orcid.org/0000-0003-4475-8887 Okpala Charles Odilichukwu R. 6
1 Department of Food Science and Engineering, South China University of Technology , Guangzhou, Guangdong , China
2 Department of Food Science and Technology, Federal University of Technology , Owerri, Imo State , Nigeria
3 Department of Food Technology, Nutrition, and Consumer Sciences, Sokoine University of Agriculture , Chuo Kikuu, Morogoro , Tanzania
4 Department of Food Science, University of Ibadan , Ibadan , Nigeria
5 Department of Biochemistry, Federal University of Agriculture , Abeokuta, Ogun , Nigeria
6 Faculty of Biotechnology and Food Science, Wroclaw University of Environmental and Life Sciences , Wroclaw , Poland
Uversky Vladimir
Electronic publication date: 2021 Sep 1
Publication date: 2021
Volume: 9
Electronic Location ID: e11940
Received 2021 Mar 26; Accepted 2021 Jul 19
Copyright: © 2021 Ofoedu et al.
Copyright year: 2021
Copyright holder: Ofoedu et al.
License: This is an open access article distributed under the terms of the Creative Commons Attribution License, which permits unrestricted use, distribution, reproduction and adaptation in any medium and for any purpose provided that it is properly attributed. For attribution, the original author(s), title, publication source (PeerJ) and either DOI or URL of the article must be cited.
License URL: https://creativecommons.org/licenses/by/4.0/

Keywords: Vitamin absorption, Vitamin transport, Micronutrient, Physiological function, Animal-based, Plant-based

Funding: Chinese Scholarship Council (CSC) and South China University of Technology, Guangzhou, Guangdong, China Wroclaw University of Environmental and Life Sciences, Wroclaw, Poland Wrocław University of Environmental and Life Sciences and co-financed by the European Social Fund POWR.03.05.00-00-Z062/18 Chigozie E. Ofoedu, Victory S. Igwe, James S. Chacha, and Adedoyin O. Agunbiade received financial support from the Chinese Scholarship Council (CSC) and South China University of Technology, Guangzhou, Guangdong, China. Maciej Korus, Joncer Naibaho, and Charles Odilichukwu R. Okpala received financial support from the Wroclaw University of Environmental and Life Sciences, Wroclaw, Poland. This publication was financed by the project UPWR 2.0: international and interdisciplinary programme of development of Wrocław University of Environmental and Life Sciences, co-financed by the European Social Fund under the Operational Program Knowledge Education Development, under contract No. POWR.03.05.00-00-Z062/18 of June 4, 2019. There was no additional funding received for this study. The funders had no role in study design, data collection and analysis, decision to publish, or preparation of the manuscript.

==============================
The significant attention gained by food-sourced vitamins has provided insights into numerous current researches; for instance, the potential reversal of epigenetic age using a diet and lifestyle intervention, the balance between food and dietary supplements in the general population, the role of diet and food intake in age-related macular degeneration, and the association of dietary supplement use, nutrient intake and mortality among adults. As relevant literature about food-sourced vitamin increases, continuous synthesis is warranted. To supplement existing information, this perspective review discussed food-sourced vitamins for consumer diet and health needs, scoping from vitamin absorption, metabolic functions, utilization, to balancing nutritional requirements. Relevant literatures were identified through a search of databases like Google Scholar, Web of Science, the Interscience Online Library, ScienceDirect, and PubMed. We demonstrated that vitamins whether from plant- and animal-based sources are prerequisites for the metabolic functions of the human body. The fat- and water-soluble classification of vitamins remains consistent with their respective absorption and dissolution potentials, underpinned by numerous physiological functions. Vitamins, largely absorbed in the small intestine, have their bioavailability dependent on the food composition, its associated interactions, as well as alignment with their metabolic functions, which involves antioxidants, coenzymes, electron acceptor/donor, and hormones. Moreover, vitamin deficiencies, in every form, pose a serious threat to human health. Vitamin toxicities remain rare, but can still occur mainly from supplementation, although it appears much less in water-soluble vitamins of which some excesses get readily removed by the human body, different from the fat-soluble ones that are stored in tissues and organs. Besides discussions of absorption, transport, and cellular uptake of vitamins, this perspective review also included approaches to meeting vitamin requirements and therapeutic strategies against micronutrient deficiency and COVID-19. We have also attempted on how to strike the balance between food-sourced vitamins and dietary supplements.

Introduction

Nutrients are a prerequisite to human life, and among them include vitamins that play several important physiological roles (Combs, 2008). Being low molecular weight organic compounds distinct from macronutrients such as carbohydrates, proteins, and fats, vitamins are micronutrients the human body requires for optimal cell development/growth, as well as countless metabolic functions/processes (Combs, 2008; Goncalves et al., 2015). Besides assisting in the synthesis of nervous system chemicals, blood vessel formation, hormones, and genetic materials, it is widely understood that vitamins can, when combined with proteins, serve as a catalyst to generate metabolically active enzymes necessary for life processes (Cooper, 2000; Bhagavan & Ha, 2015). Though the human body needs vitamins to work properly, most vitamins appear not synthesised within the body, at least not in sufficient amounts to meet our daily needs (SAL, 2017; Kumanyika & Oria, 2017), and therefore must be obtained from the diet. Compared to other classes of such nutrients like carbohydrate, protein, and fat, the breakdown (catabolism) of vitamins neither provides significant energy nor structural functions to the body. Instead, very small quantities (micrograms or milligrams per day) are required to perform the highly specific and unique functions of health maintenance and metabolic integrity (Bender, 2003). This, therefore, distinguishes vitamins from fatty acids and amino acids, which are nutrients required in larger amounts, and also from inorganic minerals and trace elements. Briefly, vitamins are distinguishably grouped into fat-soluble vitamins (A, D, E, and K) and water-soluble vitamins (B-complex and C) based on their absorption potential in either fat (non-polar medium) or water (polar medium), in addition to the region of their physiological activity (Zhao et al., 2019). The magnitude of their dissolution is influenced by some intrinsic and extrinsic factors which would be discussed later in this perspective review. Generally, the fat-soluble are well-known as lipophilic (oil-loving) organic compounds while the water-soluble vitamins are regarded as the hydrophilic (water-loving) organic compounds.

However, besides being a disparate group, vitamins nutritionally form a cohesive group of organic compounds that are neither chemically identical nor able to provide similar metabolic functions (Bender, 2003). Vitamins as micronutrients have assumed outstanding public health importance. As a result, there continues to be an increase in evidence regarding their physiological function and associated health implications of their deficiencies in diets. For instance, as vitamin A deficiency has been implicated to be the leading cause of blindness and high risk of death from common illnesses such as diarrhoea in children, it has been classified by the World Health Organization as a public health problem affecting children in South Asia and sub-Saharan Africa (UNICEF, 2019). Unlike macronutrient (carbohydrate and protein) deficiency, the health implications of micronutrient deficiencies are not always visible, hence the term hidden hunger. It is well recognized that malnutrition in every aspect poses a great threat to public health, as the World Health Organization estimates the number of people with micronutrient deficiency in the world to be over two billion (UNICEF, 2019). The prevalence of the most common vitamin deficiency globally is briefly shown in Table 1. Given this, understanding the significance as well as the magnitude of (public health) the severity of this micronutrient’s malnutrition, and providing adequate control and preventive measures is very crucial (FAO/WHO, 2004). No single food source contains all the vitamins, and importantly, inadequate or suboptimal consumption of vitamins result in deficiencies. This implies that different foods are needed to meet the vitamin requirement of the human body (SAL, 2017). Notably, the occurrence of micronutrient (vitamin) malnutrition in an individual, could be an indication of vitamin deficiency, attributable not only to poor consumption of vitamin-rich diets but also due to factors influencing vitamin absorption. Besides inadequate ingestion of vitamin-rich foods, some of the factors that can cause vitamin deficiency as a result of impairment of intestinal vitamin absorption are genetic disorders in transport molecules, intestinal disease, interactions with certain drugs and some food components, and excessive alcohol consumption (Said, 2004; Iqbal & Hussain, 2009; Bowen, 2018).

Table 1 Prevalence of the most common global vitamin deficiency.

Vitamin	Prevalence of deficiency	Sufficiency	Hypovitaminosis	Deficiency	Reference(s)	
B12	>50%	>221 pmol/L	≤148–221 pmol/L	<148 pmol/L	WHO (2008); Guney, Yikilmaaz & Dilek (2016); Allen et al. (2018)	
A	13%	>0.83 µmol/L	≤0.71–0.83 µmol/L	<0.70 µmol/L	WHO (2011); Akhtar et al. (2013)	
B9	>20%	>30 nmol/L	–	<10 nmol/L	WHO (2008); Bailey et al. (2015); Rogers et al. (2018)	
D	>20%	50–100 nmol/L	≤30–50 nmol/L	<30 nmol/L	Roth et al. (2018); Amrein et al. (2020)	
C	>1.4%	50–75 µmol/L	≤23–28 µmol/L	<11 µmol/L	Rowe & Carr (2020)	
E		23–73 µmol/L	–	<17 µmol/L	Institute of Medicine (2000); Charbek (2014)	
Note:

Values are the amount of blood serum vitamin.

Understanding the key mechanisms involved in vitamin absorption has been among the key challenges confronting researchers in this food-related discipline. For instance, some researchers of previous studies provided evidence suggesting that some fat-soluble vitamins were absorbed by passive diffusion while other vitamins of the same class were absorbed through carrier-dependent proteins (Goncalves et al., 2015). Interestingly, recent studies have shown that the mechanism of vitamin absorption, as well as transportation of absorbed vitamins, are somewhat compounded than earlier stated, as their mode of absorption appears to be concentration-dependent. For example, a high concentration of these fat-soluble vitamins is absorbed through passive diffusion while protein-mediated transport occurs at dietary doses (Reboul & Borel, 2011). On the other hand, particularly the water-soluble ones, it is well understood that vitamin absorption does occur via diffusion.

Briefly, the absorption of vitamins begins with chewing of ingested food, where enzymes start to play their role from the mouth, together with continued churning and mixing with other chemical constituents in the stomach. The churned food makes contact with the small intestine, which is the nutrient absorption center in the body. Further, the water-soluble vitamins diffuse through the intestinal walls into the bloodstream while the fat-soluble vitamins are emulsified and packaged in lipid-rich mixed micelles containing fatty acids, bile salt, and phospholipids. Subsequently, fat-soluble vitamins would be moved through the brush-like projections (villi) and absorbed into the lymphatic circulation where they are transported to tissues, target cells, or organs (liver) (El-Kattan & Varma, 2012; Khammissa et al., 2018). As relevant literature regarding food-sourced vitamins increases by the day, there is a need for continuous synthesis to update existing information. To supplement existing information, this perspective review discussed food-sourced vitamins for consumer diet and health needs, scoping from vitamin absorption, metabolic functions, utilization, to balancing nutritional requirements. For emphasis, this perspective review has been constructed as follows: (a) classification of vitamins; (b) sources of vitamins; (c) metabolic functions of vitamins; (d) absorption of vitamins; (e) vitamin transport; (f) characteristic interactions associated with food-sourced vitamins (g) vitamin deficiencies and toxicities; (h) approaches to meeting vitamin requirements; (i) vitamins as a therapeutic tool against micronutrient deficiency and COVID-19; as well as (j) establishing a balance between food-sourced vitamins and dietary supplements. This work intends to disseminate a succinct summary of existing information about key vitamin research studies and at the same time, revisit the state-of-the-art. In considering its exceptional public health importance, the physiological functions, and associated health implications of respective vitamin deficiencies, the need for the continuous literature synthesis on the severity (of the diverse vitamin deficiency) around the globe, therefore, makes this review very fitting. Following this, the current perspective review is essential, not only for the general public, but also for nutritionists, dietitians, food scientists, public health, and health-related specialists, together with their respective organizations, to understand better the mechanisms of vitamin transport and utilization, including the rationale for the impairment of vitamin absorption.

Survey methodology

The formulating of research questions was the very step in actualising this perspective review. We made effort to ensure the research questions were directly relevant to the intended objective, and at the same time, considered the intended/target audience. After this, a search strategy was developed by consulting and collating published papers and their references from selected scientific databases, including Google Scholar, ISI Web of Science, ResearchGate, Interscience Online Library, ScienceDirect, Semantic Scholar, and PubMed/Medlin. Each database was searched using keywords and search terms such as vitamin absorption, mechanism of vitamin absorption, utilization of vitamin, transportation of vitamin, and cellular uptake of the vitamin. Further, the inclusion and exclusion criteria were established, which through the help of search strategy helped to determine the research articles that are appropriate, relevant, and specific to the objective of this perspective review.

Primary research studies reporting the fundamental aspects of vitamin in nutrition and health were included. Specifically, studies were selected if they had evaluated vitamin and their associated body functions, vitamin and food component interaction, as well as vitamin requirement in human nutrition. Articles reporting food-sourced vitamins especially from the plant and animal origin contexts were equally included because we considered them relevant for this review. Excluded studies were those that evaluated and reported other sources of vitamins such as vitamins exclusively produced from microorganisms, vitamins as nutritional supplements, vitamin-like compounds, and vitamin mimic. It is important to reiterate here that all the publications in languages other than English were also excluded. Additionally, we applied the discretion to strike a balance between the year of publication and the relevance of the information to the objective of this review. Next, the information gathered from the selected literature/studies were analysed. Essentially, all the information deemed relevant were incorporated, which helped to expatiate the discourse of this conducted perspective review.

Discussion of literature synthesis

Classifications of vitamins

In human nutrition, the traditional and widely accepted means of classifying the well-known thirteen vitamins have been based on their solubility, which includes: fat-soluble (hydrophobic) and water-soluble (hydrophilic) vitamins. The flow diagram showing the classifications of vitamins is shown in Fig. 1. This classification of vitamins is basically dependent on their absorption potential either in fat or water (Bhagavan & Ha, 2015), together with where they act in the body (Delaney & Barke, 2017). The degree of vitamin dissolution in fat or water varies and is influenced by intrinsic and extrinsic factors. The fat-soluble vitamins are oily hydrophobic (lipophilic) organic compounds, usually not excreted out of the body, but rather stored in the body (liver), which include vitamins A, D, E, and K. In contrast, the water-soluble vitamins are hydrophilic (lipophobic) organic compounds not stored in the body and easily excreted through urine. Water-soluble vitamins are, therefore, required daily in small amounts, which include vitamin C (ascorbic acid) and the members of the vitamin B group, namely thiamine (vitamin B1), riboflavin (vitamin B2), niacin (vitamin B3), pantothenic acid (vitamin B5), pyridoxine (vitamin B6), biotin (vitamin B7), folate (vitamin B9), and cobalamin (vitamin B12) (Ball, 2004). Further, it is well-known that almost all vitamins are considered essential for their role in physiological processes and must be obtained from food, but some vitamins that can be synthesized endogenously in the body are not technically considered “essential” and they include vitamins D, B7, and K. The chemical structure of all the vitamins are shown in Fig. 2.

Figure 1 Classifications of vitamins.

Figure 2 Chemical structures of vitamins.

(A) Vitamin A (Retinol); (B) Vitamin D (Ergocalciferol); (C) Vitamin E (α-Tocopherol); (D) Vitamin K - (i) Phylloquinone (ii.) Menaquinone (iii.) Menadione; (E) Vitamin B1 (Thiamine); (F) Vitamin B2 (Riboflavin); (G) Vitamin B3 (Niacin); (H) Vitamin B5 (Pantothenic acid); (I) Vitamin B6 (Pyridoxine); J: Vitamin B7 (Biotin); K: Vitamin B9 (Folate); (L) Vitamin B12 (Cobalamin); (M) Vitamin C (Ascorbic acid).

(a) Fat-soluble vitamin group

(i) Vitamin A

Generically, vitamin A is a group of essential fat-soluble micronutrients of identical compounds known as retinoid (Delaney & Barke, 2017). Vitamin A is categorised in dietary form as either provitamin A or previtamin A. The provitamin A is exclusively synthesised by plants and present in the diet in foods of plant origin. Provitamin A is mainly composed of carotenoids consisting of β-cryptoxanthin, α-carotene, and β-carotene (Ross & Harrison, 2014). Carotenoids are pigments produced by plants that give plants their range of colours, such as red, yellow, and orange. The body cells can utilize and convert (transform) carotenoids (β-cryptoxanthin, α-carotene, and β-carotene) to vitamin A (Newman, 2017), given the β-carotene conversion rate to vitamin A as 12:1 and 24:1 for β-cryptoxanthin (Scott & Weir, 1994; Goncalves et al., 2015).

On the other hand, previtamin A, in the form of vitamin A, is synthesized by animals and derived from diets in foods of animal origin. This type of vitamin A is specifically active in the body as retinol (alcoholic form), retinal (aldehyde form), and retinoic acid (acidic form) (Delaney & Barke, 2017; SAL, 2017). However, retinol found in foods is chemically unstable though the extent of its occurrence in foods and tissues is insignificant (Bender, 2003). Each form of vitamin A performs a unique function. Retinol assists in reproduction and it is vitamin A’s major form of transport, retinal serves as an intermediate in the transformation (conversion) of retinol to retinoic acid and it’s also crucial in the vision system and pigment, while retinoic acid acts like a hormone responsible for the growth, development, and maintenance of epithelial cellular integrity, regulation of cell differentiation and immune function (FAO/WHO, 2004; SAL, 2017), as well as modulates gene expression (Bender, 2003). Notably, retinal and retinoic acid are derivatives of retinol.

(ii) Vitamin D

Vitamin D denotes a group of non-essential micronutrients of fat-soluble compounds derived from cholesterol. The biologically active forms of vitamin D relevant to human is synthesized from provitamin D2 (ergocalciferol) and provitamin D3 (cholecalciferol) (Zheng & Teegarden, 2014). The body (skin) synthesizes vitamin D when exposed to the ultraviolet B (UVB) radiation of the sunlight. Thus, the reason for being non-essential micronutrients since it can be produced by the body provided the season is right and duration of exposure in the sun (Delaney & Barke, 2017), whereas vitamin D2 is obtained from foods of plant origin, specifically plant steroid, ergosterol. Both vitamins D2 and D3 are not active until activated by the body. As a result of its endogenous production, vitamin D in its active form is classified as a hormone, as it targets the bones, intestines, and kidneys (Zheng & Teegarden, 2014; SAL, 2017). As a steroid prohormone (Holick, 2004; Silva & Furlanetto, 2017), vitamin D is of increasing global concern probably due to its significance in bone health and other important systematic functions (Chung et al., 2009; Verstuyf et al., 2010; Martini et al., 2013). Vitamin D is believed to have a multitude of (biological) functions. For instance, its function in regulating the immune system and cells, increasing the intestinal absorption of calcium and phosphorus, helping the bones to grow stronger and denser (builds and maintains strong bones) as they absorb and deposit calcium and phosphorus, including the mineralization of bones mediated through osteoblasts (SAL, 2017) are among the key functions associated with vitamin D.

(iii) Vitamin E

Vitamin E exists in many forms of which the most active form relevant to meet human requirements is alpha (α)-tocopherol (Delaney & Barke, 2017). Unlike other vitamins with specific tasks such as being a cofactor, hormones, or participating in cell metabolism, the vitamin E is an unsaturated fat-soluble organic compound uniquely positioned to act as an antioxidant and free radical scavenger (i.e. inhibits free radical chain reactions that cause oxidative stress) against lipid oxidation (Hs et al., 2019). As an essential micronutrient, vitamin E is obtained in diets of foods from plant origin as it is synthesized by plants and other photosynthetic organisms only (DellaPenna, 2005; Reboul, 2017). Like other fat-soluble vitamins, the dietary fat is needed for the absorption of vitamin E since its transportation to the liver is dependent on being packaged into lipid-rich chylomicrons in the intestinal cells (Delaney & Barke, 2017).

(iv) Vitamin K

Vitamin K is an essential fat-soluble organic compound (micronutrient) that belong to the isoprenoid quinones (Shearer & Newman, 2014) having cofactor activity for ℽ-glutamyl carboxylase (Ball, 2004). Believed to exist in two forms, vitamin K is synthesized by plants and bacteria. In plants, vitamin K1 known as phylloquinone with phytyl side chain is, on one hand, the only vital molecular form of vitamin K. On the other hand, vitamin K2 known as menaquinones with unsaturated 5-carbon (prenyl) units is synthesized by bacteria (FAO/WHO, 2004). Essentially, both vitamins K1 and K2 are the main circulating and storage forms of vitamin K in humans (Ottaway, 1993). Additionally, vitamin K3 (menadione) is one of the artificially produced forms of vitamin K that does not occur naturally. It is in the form of a yellowish synthetic crystalline substance, usually converted into an active form of vitamin K2 in the body of an animal (Jan et al., 2015; Drugbank, 2021). As an essential micronutrient, vitamin K is not synthesized by humans and therefore should be obtained from plant-based diets. Vitamin K is critical to good health as it plays a significant function in bone health or formation and blood coagulation known as clotting (Combs, 2008). In addition, vitamin K is involved in bone protein metabolism, especially osteocalcin. Notably, minerals cannot interact with osteocalcin that normally forms bones without vitamin K activity (SAL, 2017).

(b) Water-soluble vitamins group

(i) Vitamin B1

This is the first member of B-complex vitamins known as thiamine, and a sulfur-containing water-soluble type with pyrimidine and thiazole rings, linked by a methylene bridge (Bender, 2003). Thiamine acts as a coenzyme (thiamine pyrophosphate) in a variety of important metabolic reactions as it plays a significant role in energy metabolism, catabolizing carbohydrates, proteins, and lipids into energy (FAO/WHO, 2004). It is also essential in muscle and nerve activity (SAL, 2017), as the glucose that is broken down by thiamine is needed as an energy source for these activities. In addition, as a key player in the synthesis of neurotransmitters, it is also required for ribonucleic acid (RNA), deoxyribonucleic acid (DNA), and adenosine triphosphate (ATP) synthesis (Delaney & Barke, 2017).

Thiamine is very sensitive to decomposition by many factors such as heat, neutral and alkaline pH conditions, ionizing radiation (light), sulphite during processing, and oxidation (Combs, 2008). Thiamine is obtained by humans from diets as it is mostly found in foods though in a relatively low amount (Said, 2004; Combs, 2008). It is obtained from bacteria sources, especially the normal microflora in the intestine and usually, it is also absorbed in that same gut region (Said, 2004). Thiamine’s occurrence in the diet is in the phosphorylated form, which is converted to free thiamine by phosphates before its intestinal absorption. Additionally, most of the thiamine produced in the large intestine by bacteria exist in its free absorbable form but the total body thiamine requirement relative to the contributions from bacteria source is yet unclear (Said, 2011).

(ii) Vitamin B2

Vitamin B2 is a member of water-soluble vitamins known as riboflavin. It supports cellular antioxidant protection and plays a key role in the metabolic pathway of carbohydrates, lipids, and proteins (Combs, 2008; Said, 2011). Riboflavin also discharges its functions by acting as a precursor to the coenzymes, flavin mononucleotide (FMN) and flavin adenine dinucleotide (FAD) which are historically referred to as nucleotides. They are involved in a wide range of oxidation-reduction reactions in the intermediary cellular metabolism. As a coenzyme, it is an essential component of flavoproteins (flavoenzymes), which assist in electron transfer in the electron transport chain (ETC) (Said, 2011). It is believed that the coenzymes of other B-vitamins (vitamin B6, niacin, and folate) rely on the actions of flavoproteins for proper functioning and conversion into their active coenzyme forms (Said, 2011).

The riboflavin widely found in foods is exclusively bound to proteins in the form of FMN and FAD (Decker, 1993; Bates, 1997; Combs, 2008). Besides being sensitive or easily destroyed by ultraviolet light, vitamin B2 also acts as auxiliary molecules to help ensure the proper functioning of proteins in the body (Avrutin, 2020). Like in other vitamins, riboflavin is obtained from diets of plant-based and animal-based, which are processed and absorbed in the small intestine (Said, 2011; Pinto & Rivlin, 2014; SAL, 2017). Additionally, it can be obtained from bacteria in the large intestine, where the vitamin is synthesized and absorbed in the same gut region (Said, 2011).

(iii) Vitamin B3

This is another member of the water-soluble vitamin known as niacin. Niacin exists in two forms, namely; nicotinic acid and nicotinamide with its biologically active forms of nucleotides as nicotinamide-adenine dinucleotide (NAD) and nicotinamide-adenine dinucleotide phosphate (NADP) through which the vitamin participates virtually in all aspects of metabolism (Combs, 2008). As coenzymes (NAD and NADP), niacin participates in energy-transfer reactions, specifically in the metabolic reactions of carbohydrates, lipids, proteins, and alcohols. Also, niacin aids in nervous system function and digestion, participates in oxidation-reduction metabolic reactions, protects against neurological degeneration, and has lipid-lowering ability (Trueblood et al., 2000; SAL, 2017; Delaney & Barke, 2017; Avrutin, 2020).

The implication of oxidative stress in a variety of chronic or neurodegenerative diseases cannot be overemphasized (Ilkhani, Hosseini & Saedisomeolia, 2016; Ofoedu et al., 2021). As a neglected potential antioxidant micronutrient, the active form (NAD/NADP) or reduced form (NADH/NADPH) of niacin have been associated with antioxidant properties via glutathione redox cycle, which tends to combat oxidative stress, specifically, lipid peroxidation and reperfusion oxidative injury (Ilkhani, Hosseini & Saedisomeolia, 2016; Fricker et al., 2018; Gasperi et al., 2019). Additionally, the lipid-lowering ability of niacin (e.g., reduction in triglycerides, low-density lipoprotein (LDL), very-low-density lipoprotein (VLDL), and total cholesterol) varies according to the mode of usage, either singly or combined with other medications, for instance, hydroxymethyl glutaryl coenzyme A reductase inhibitors (Djadjo, 2021). Niacin’s mechanism of action for its antilipolytic effect has been hypothesized to be mediated through nicotinic acid receptors, which can enhance intracellular degradation of Apolipoprotein B (ApoB) (that contains lipoproteins such as LDL and VLDL), by inhibiting the synthesis of triglycerides (Florentin et al., 2011; Kei, Liberopoulos & Elisaf, 2011; Djadjo, 2021).

Interestingly, niacin’s coenzymes are involved in electron transfer in the electron transport chain (ETC) during cellular respiration to make ATP and are also essential in the synthesis of cholesterol and build-up of fatty acids (Said, 2011). Niacin can be biosynthesized in the body from amino acid tryptophan (SAL, 2017; Delaney & Barke, 2017). This synthesis takes place only after tryptophan has met its nutritional requirement in the body. However, studies have shown that there is little or no association between food high in tryptophan and niacin deficiency (Delaney & Barke, 2017). Niacin in foods is very stable to normal storage conditions and other food preparation and cooking methods, for example, moist heat (Combs, 2008). The vitamin is found in various foods and predominantly occurs in bound forms. For instance, in plants, it is present mostly as protein-bound nicotinic acid and as nicotinamide in animal tissues (Combs, 2008).

(iv) Vitamin B5

Vitamin B5 is a water-soluble vitamin known as pantothenic acid which has a central role in energy-yielding metabolism as a component of coenzyme A (main carbon molecule carrier in a cell) moiety. It is a cofactor that carries acyl group in many enzymatic processes, acyl transfer reactions (Bender, 2003), and in the biosynthesis of fatty acids (FAO/WHO, 2004; Bender, 2003), haemoglobin, steroid hormones, and neurotransmitters (SAL, 2017). Further, it is essential for normalizing the blood lipid profiles, tissue repair, and maintenance, as well as in a wound, and lesion healing (SAL, 2017). Pantothenic acid can be obtained from diets of food of plant and animal origin, as well as bacterial sources, from natural microflora of the large intestine (Said, 2011). The vitamin is ubiquitously distributed in many foods, which is why its deficiency is rare. Interestingly, it occurs mainly in the bound forms of CoA, CoA esters, and acyl-carrier proteins (Combs, 2008).

(v) Vitamin B6

Vitamin B6 is an essential water-soluble vitamin occurring in three natural forms, namely; pyridoxine (alcohol form), pyridoxamine (amine form), and pyridoxal (aldehyde form). Vitamin B6 is essential for many biological functions such as the metabolism of three macronutrients (carbohydrate, protein, and lipid), synthesis of neurotransmitters, and in a variety of elimination and replacement reactions (Bender, 2003; SAL, 2017). Besides being involved in the formation of red blood cells or haemoglobin synthesis, vitamin B6 coenzymes (pyridoxal phosphate) plays a significant role in multiple metabolic reactions, giving the vitamin importance in such diverse areas as growth, cognitive development, depression, immune function, fatigue, and steroid hormone activity (Combs, 2008).

Vitamin B6 has been closely linked with the function of the nervous system as a result of monoamine neurotransmitters generated via the amino acid decarboxylase reactions (Dakshinamurti & Dakshinamurti, 2014). The coenzyme of vitamin B6 plays a key role in a valuable protein and urea metabolism (SAL, 2017) by the conversion of essential amino acid methionine into cysteine (non-essential amino acid), as well as in the conversion of amino acid tryptophan to niacin or the neurotransmitter serotonin (SAL, 2017). Vitamin B6 is fairly widespread in plant-based and animal-based foods, thus its deficiency is not prevalent (Combs, 2008). Interestingly, under acidic conditions, vitamin B6 is stable but fairly unstable in neutral and alkaline conditions especially when exposed to heat or light.

(vi) Vitamin B7

Vitamin B7 is a water-soluble vitamin also known as biotin or vitamin H. Biotin is very important in metabolic reactions such as gluconeogenesis, lipogenesis, and breakdown (catabolism) of branched-chain amino acids (Bender, 2003). Biotin is essential as a coenzyme that transfers carbon dioxide (CO2), which is crucial in the catabolism of carbohydrates, proteins, and fats into energy (SAL, 2017). Additionally, it functions as a coenzyme involved in many cellular metabolisms and the synthesis of some non-essential amino acids. Biotin is absorbed in the gut by the specific action of an enzyme, biotinidase, which releases biotin from protein (Delaney & Barke, 2017). Similarly, it is also a key player in the regulation of oncogenes expression and the cellular level of the second messenger cGMP (Scheerger & Zempleni, 2003; Said, 2011). Biotin is obtained from different varieties of foods. It can also be synthesized by bacteria in the colon, but in an insignificant amount (Combs, 2008; Delaney & Barke, 2017). However, biotin in food is easily destroyed by heat and unstable in oxidizing conditions especially under conditions supporting lipid peroxidation (Dakshinamurti, 2005; Combs, 2008).

(vii) Vitamin B9

Vitamin B9 is a family of water-soluble vitamins that function in multiple forms of reactions involved in coenzymes, transferring one carbon for the synthesis of DNA and in energy metabolism (Bailey et al., 2014) which exists in two forms, namely; folate and folic acid. Vitamin B9 exists naturally as folate which has the basic molecule of 5,6,7,8-tetrahydropteroylglutamate also referred to as pteroylglutamic acid or tetrahydrofolate (THF) (Shane, 2009). The addition of glutamate residue via ℽ-peptide linkage assures that the monoglutamate form of the vitamin which contains one glutamic acid molecule is converted to a polyglutamate chain (varies from 5 to 8 side chain). On the other hand, folic acid is a synthetic fully oxidized monoglutamate form of the vitamin that is used commercially in fortified or enriched foods and supplements (Bailey et al., 2014; SAL, 2017). Folic acid (Pteroyl monoglutamate) is the most stable form of folate. In addition to folate’s action as a coenzyme, it is essential for the growth, development, and specialization of cells of the central nervous system (Delaney & Barke, 2017), and brain development and function. Similarly, it synthesizes thymidine and purine as well as participates in several amino acid metabolisms, for example, homocysteine (Said, 2011).

Interestingly, folate and vitamin B12 collaborate in the transfer of a one-carbon compound which is the methyl group. For instance, folate’s coenzyme (THF) donates its carbon compound from a methyl group to vitamin B12, thereby transforming the latter into an active coenzyme and as a result, catalyzes the conversion of homocysteine to methionine (SAL, 2017). In other words, in the absence of vitamin B12, late in its methyl form becomes enclosed in cells, and unavailable to support cell growth. Folate is widely distributed in foods of plants and animal origin, although the latter is not quite a good source of folate. In most foods containing folate other than folic acid, the former is easily oxidized due to its instability to oxidation under aerobic conditions of storage and processing (Combs, 2008).

(viii) Vitamin B12

Vitamin B12 is an essential water-soluble vitamin known as cobalamin, making it the only vitamin that contains metal, cobalt (Delaney & Barke, 2017). Cobalamins are corrinoids with a dimethylbenzimidazole nucleotide attached to the D ring and chelating the central cobalt atom. In nature, vitamin B12 occurs in two forms, namely: hydroxocobalamin and aquacobalamin, where hydroxyl and water groups are respectively attached to the cobalt. However, the synthetic form of vitamin B12 is found in enriched/fortified foods, as well as in such supplements as cyanocobalamin, which has cyanide attached to the cobalt. Noteworthily, aquacobalamin, hydroxocobalamin, and cyanocobalamin are enzymatically activated in all mammalian cells to the methyl or deoxy adenosylcobalamin (FAO/WHO, 2004). In mammalian cells, methionine synthase and methyl malonyl coenzyme (CoA) mutase are the two vitamin B12-dependent enzymes that are necessary for its metabolic processes (Scott & Rodriquez, 2000; FAO/WHO, 2004).

As an important coenzyme, vitamin B12 is necessary for the breakdown of protein and fat, synthesis of haemoglobin, activation of folate’s coenzyme function (Delaney & Barke, 2017), in cellular division and growth (Combs, 2008), and participates in the conversion of homocysteine to methionine (SAL, 2017). Further, in dietary terms vitamin B12 is not derived from plant-based foods; therefore, humans and animals that consume strictly vegetarian diets are very likely to have the suboptimal intake of vitamin B12 except where the plant-based foods have been fortified or enriched (SAL, 2017). On the other hand, vitamin B12 is found in the tissues of animals and can also be synthesized by bacteria (FAO/WHO, 2004; Combs, 2008).

(ix) Vitamin C

Vitamin C is a family of water-soluble compounds also known as ascorbate in dietary terms but occurs in two forms, namely; ascorbic acid (reduced form) and dehydro-L-ascorbic acid (DHAA) (oxidized form) (Said, 2011). Chemically, it is a six-carbon lactone that is synthesized from glucose by many animals (FAO/WHO, 2004) but cannot be synthesized by humans because of a single enzyme deficiency (Combs, 2008). Whether or not it is synthesized, ascorbic acid is an important vitamin for several physiological functions. As a cofactor in multiple integral metabolic reactions, vitamin C is essential in the synthesis of collagen, defense against infection and inflammation, as well as maintaining metal ions, for example, iron and copper, in their reduced states (FAO/WHO, 2004; Delaney & Barke, 2017). Vitamin C plays an important role as a catalyst and redox cofactor in a wide variety of biochemical processes and reactions (Johnston, Steinberg & Rucker, 2014).

Besides acting as a cofactor in helping specific enzymes do their functions properly, ascorbic acid has a protective role of acting as an antioxidant (reducing agent or electron donor) in tissues and plasma against free radicals (FAO/WHO, 2004; Combs, 2008; SAL, 2017). The ability of vitamin C to freely donate electrons makes it an effective scavenger against reactive oxygen species and reactive nitrogen species (Delaney & Barke, 2017), as well as in arresting lipid oxidation. Additionally, it supports the recycling and regeneration of α-tocopherol (vitamin E) (Combs, 2008). In addition to vitamin C’s antioxidant capacity, it acts as an electron donor to several enzymes (Combs, 2008), necessary for signalling molecules in the brain, some hormones, and amino acids (Delaney & Barke, 2017). It is also essential in the growth and repair of tissues in all body parts.

Metabolic functions and sources of vitamins

Indeed, vitamins play key roles in the physiological processes of our bodies and perform unique metabolic functions that are dependent on their chemical reactivity and cellular or tissue distribution. The various functions of vitamins form the basis for their relevance in health and nutrition. Generally, the metabolic functions of vitamins are specifically categorized into antioxidants (vitamins A and E), coenzymes (vitamins A, K, C, B1, B2, B3, B5, B6, B7, B9, and B12), hormones (vitamins A and D), and electron donors/acceptors (K, C, B5, B3, and B2), and gene transcription elements (Combs, 2008). While most vitamins require metabolic activation or conversion to perform their physiological role in biological systems, few vitamins function without any kind of metabolic conversion or association with an enzyme. The metabolic modification of food-sourced vitamins into active forms utilized during cellular metabolism may cause significant transformation on the chemical properties and/or structure of the vitamin which can greatly affect their nutritional potency. In addition, summarized in Table 2 are the specific metabolic functions of vitamins when considered individually.

Table 2 Metabolic functions of vitamins.

Vitamins	Functions	
A	Promotes good vision, produces visual pigments in the retina of the eye, and participates in epithelial cell differentiation	
D	Regulates bone metabolism, mineralization of bones via osteoblasts, and increases calcium and phosphate absorption in the intestine	
E	Possess membrane antioxidant capacity	
K	Important for blood clotting and calcium metabolism for bone health	
C	Acts as an antioxidant in tissue and plasma, promotes growth and repair of tissues, essential for the synthesis of collagen and carnitine	
B1	Vital for energy metabolism, nerve and muscle activity, forms important coenzymes for decarboxylations of 2-keto-acids and transketolations	
B2	Forms an important coenzyme in oxidation-reduction reactions of fatty acids and tricarboxylic acid (TCA) cycle, essential for energy metabolism and synthesis of neurotransmitters	
B3	Essential for energy metabolism and neurological processes, an important coenzyme for several dehydrogenases	
B5	Essential for wound healing and normalizing blood lipid profile, an important coenzyme in fatty acid metabolism	
B6	Important for nerve activity, acts as coenzymes in amino acid metabolism, fundamental in the synthesis of neurotransmitters, DNA and haemoglobin	
B7	Important coenzyme for carboxylation and in lipid metabolism, essential for hair, skin and nail regeneration	
B9	Vital coenzyme in single-carbon metabolism and synthesis of DNA, necessary for red blood cell production and neural tube formation	
B12	Necessary for folate coenzyme function, red blood cell formation, and absorption of iron, calcium and vitamin A, essential in nerve activity and synthesis of neurotransmitters	
Note:

Source: Adapted from Combs (2008).

Vitamins exist naturally in different forms. Some are synthesized by microorganisms (intestinal bacteria) in the gut region, for example, vitamins K2, B1, B9, and B12 (Gorbach, 1996; Hill, 1997; Uribe, Garcia-Galbis & Espinosa, 2017; Chen, Michalak & Agellon, 2018), while others are predominantly found in foods of plant origin only, animal origin only, or both plant and animal origin. Also, some vitamins are synthesized in the body in varying concentrations probably through the metabolic transformation from amino acids and/or directly produced by the skin. For example, the endogenous synthesis of niacin and vitamin D from the amino acid tryptophan and exposure of the skin to the sun respectively (Bender, 2003; Nair & Maseeh, 2012). Generally, the vitamins synthesized endogenously are not sufficient to meet daily needs, so dietary intake is required (Combs, 2008: Uribe, Garcia-Galbis & Espinosa, 2017). Moreover, the amount of vitamins in food is dependent on its source, the form it occurs naturally, and its actual distribution across various foods. However, no single food source contains all the vitamins and inadequate or suboptimal consumption of vitamins results in deficiencies. This implies that a variety of food is therefore needed to meet the vitamin requirement of the body (SAL, 2017). Table 3 summarizes the major food sources of vitamins available for the human diet.

Table 3 Major food sources of vitamins available for the human diet and recommended amount of daily adequate intake.

Vitamin	Food sources	Recommended amount (daily AI)	
A	Retinol: Fish, butter, whole milk, egg yolk, and cheese. Carotenoids: Carrots, orange-flesh sweet potatoes, orang- flesh fruits (melon, mangoes, & persimmons), green leafy vegetables (i.e. spinach, broccoli, etc.), pumpkins, and red palm oil	700–900 µg (2,333–3,000 IU)	
D	Sun ultraviolet B radiation, beef, egg, yolk, veal, mushrooms, liver, fortified cereal and dairy products, fatty fish, and fat spreads	15–20 µg (600–800 IU)	
E	Edible vegetable oils, nuts (almonds, peanuts) and nut spreads, avocado, sunflower seeds, fruits (mango, kiwifruit), meats, and leafy green vegetables (spinach, chard)	15 µg (22–33 IU)	
K	Phylloquinone: Spinach, broccoli, vegetable oils (soybean, canola, olive), green leafy vegetables (parsley, spinach, collard greens, and salad greens), and cabbage. Menaquinone: Natto (fermented soybeans, fermented cheese and curds	90–120 µg	
C	Fruits (especially citrus fruits), green leafy vegetables, potatoes, cabbage-type vegetable, liver (lamb and chicken), tomatoes, brussels sprouts, broccoli, and lettuce	75–90 mg	
B1	Citrus fruits (orange juice, strawberries, grape, pineapple, etc.), offal (liver, heart, kidneys), fish, meat (pork), asparagus, green leafy vegetables, whole grain cereals, legumes (lentils and beans), nuts, brewer’s yeast, squash, and soymilk	1.1–1.2 mg	
B2	Brewer’s yeast, meat (beef, pork), nuts, dark green leafy vegetables, eggs, milk, yoghurt, offal (liver, kidneys, heart), & whole grains cereals	1.1–1.3 mg	
B3	Dairy milk, egg, meat, fish, offal (liver), yogurt, cheese, legumes, fruits (dates, fogs, prunes, and avocados), mushrooms, nuts and others (synthesized from tryptophan)	14–16 mg	
B5	Brewer’s yeast, meat, poultry, red fish, cereals, legumes, egg yolk, milk, offals (liver, kidneys), tomatoes, potatoes, broccoli, mushroom, and green leafy vegetables	5 mg	
B6	Broccoli, fish, shellfish, chicken, milk, yogurt, & mushrooms	1.3–1.7 mg	
B7	Chicken, fish (tuna, salmon), beef, liver, nuts (peanut, walnut), chickpeas, maize and whole grain cereals, green leafy vegetables, bananas, potatoes, other meat organs, fruits, and starchy vegetables	30 µg	
B9	Nuts (walnuts, peanuts), cereals, milk, lentils, wheat germ, yeast, asparagus, beans, green leafy vegetable, and egg yolks	400 µg	
B12	Shellfish, liver, game meat (venison and rabbit), milk and milk products, some fish (salmon, trout, herring and sardines) Spinach, dark leafy greens, asparagus, beets, turnips, & mustard greens	2.4 µg	
Notes:

Source: Adapted from SAL (2017); Uribe, Garcia-Galbis & Espinosa (2017), and Harvard Medical School (2020).

Abbreviation: AI, Adequate Intake.

Absorption of vitamins

Vitamins are fundamentally absorbed in the small intestine and their bioavailability is dependent on the food composition of the diet. Basically in human nutrition, bioavailability by way of definition is the proportion of the amount of vitamin in ingested food that is absorbed in the intestines and utilized at the cellular level by the body (Ball, 2004). In this context, vitamin utilization entails the transport of absorbed vitamins to tissues, uptake by cells, and subsequent fate of the vitamin which could be converted into a form required to perform some biochemical function. The flow process of vitamin absorption is briefly described in Fig. 3. It shows how fat- and water-soluble vitamins gets to target cells when ingested from food.

Figure 3 Flow diagram showing the absorption of vitamins.

(a) Factors that influence vitamin absorption

To reiterate, absorption of a vitamin depends on how the vitamin exists relative to its chemical and physical state within the food matrix (Ball, 2004; Combs, 2008). Specifically, in foods, some vitamins occur as their coenzyme derivatives, underivatized form with a proportion linked with specific binding-proteins, and chemically bound complexes with some other components in the food matrix which cannot be absorbed or utilized until the vitamins are released from their bound forms by enzymes in the intestinal tract or tissues.

However, the complex nature of a food matrix in terms of its composition, structure, and component interactions, for example, vitamin-vitamin interaction, vitamin-mineral interaction, etc., can have a potential impact on vitamin absorption, utilization, and cellular uptake. In other words, food component interaction with vitamins may exhibit lower absorption efficiency as well as retard or enhance the absorption of vitamins compared with unbound (free) ones already ingested in the tablet form (i.e. supplements) without co-extractive. Additionally, adequate quantities of water and dietary fat are vital for the absorption of the water-soluble and fat-soluble vitamins respectively. According to a clinical trial report by Reboul, the presence of long-chain fatty acids reduced the absorption of vitamin D3 when administered at physiological doses in rats (Reboul, 2017). This report was strengthened by another trial report that monosaturated fatty acids enhanced the effectiveness of vitamin D supplementation whereas polyunsaturated reduced it. Interestingly, studies have shown that interaction amongst vitamins can influence its efficiency of absorption. A study conducted by Goncalves et al. (2015) shows that vitamin E significantly enhanced vitamin A uptake at medium and high concentrations up to 40% while vitamin D absorption was significantly reduced by vitamin E at medium (−15%) and high (−17%) concentration as well as by vitamin A at high concentration (−30%).

Furthermore, the nature (physical state) of food can also influence vitamin absorption efficiency. For example, a carotenoid present in fibrous plant material with poor digestibility has been shown to exhibit low bioavailability relative to vitamin A (Riedl et al., 1999; Lattimer & Haub, 2010). It is important to note that extrinsic factors (differing biopotencies of vitamin forms, dietary effects, losses during storage, processing, and/or cooking) and intrinsic factors (state of health and age-related differences in gastrointestinal function) may also influence the efficiency of vitamin absorption (Combs, 2008). For instance, in terms of dietary effects, the composition of diets and meals would influence the absorption of some vitamins by affecting intestinal transit time and/or the enteric formation of mixed micelles (Lattimer & Haub, 2010; Muller, Canfora & Blaak, 2018; Carreiro & Buhman, 2019; Puglisi, 2019). This was found in a study that reported poor absorption of vitamin A and pro-vitamin A carotenoids from very low-fat diets. Roodenburg et al. (2000) showed that 3 g of fat per meal would suffice for optimal absorption of some pro-vitamin A carotenoids. However, Brown et al. (2004) found that 29.5 g of fat per meal would not suffice for optimal absorption. These two studies would, therefore, suggest that other factors such as individual health status, physiological effects, and interactions between vitamins and other food components could be responsible for such variations. While it has been established that fat-soluble vitamins may not be fully absorbed from low-fat diets, the minimum amount of fat required to facilitate proper absorption of lipophilic vitamins is still not clear. Other factors that may interfere with the physiological absorption mechanism of vitamin includes alcohol and drugs, presence of gastrointestinal disorders or diseases (parasitic infections, malignant diseases, etc.), and metabolic requirement (Ball, 2004).

(b) Absorption of water-soluble vitamins

The physical and chemical properties of vitamins play an integral role in the absorption of water-soluble vitamins. The polar environment of the intestinal lumen makes the absorption of water-soluble vitamins more direct and suitable for the absorptive surface of the gut due to its hydrophilic nature (Combs, 2008). Some water-soluble vitamins such as vitamin C, pantothenic acid, niacin, folate, thiamine, vitamin B12, biotin, and vitamin B6 are absorbed into the bloodstream by the action of passive diffusion while others such as riboflavin which may not undergo simple diffusion are absorbed through a carrier-dependent mechanism as a means of overcoming concentration gradient (Halsted, 2003; Said, 2011). In other words, at high concentrations (doses), several vitamins are absorbed by simple diffusion and by specific carriers or carrier-dependent mechanisms at low concentrations (doses).

(c) Absorption of fat-soluble vitamin

As earlier indicated in this paper, the physical and chemical properties of vitamins play an integral role in their absorption. However, the hydrophobic nature of fat-soluble vitamins makes them insoluble in the aqueous environment of the alimentary canal, but instead are meant to dissolve in other lipid materials. As a result of gastric churning and mechanical actions of mastication in the upper portion of the gastrointestinal tract, the fat-soluble vitamins are solubilized in the bulk lipid phase of the emulsions formed from dietary fats (Borel, 2009). Further, due to the large size of emulsion oil droplets (e.g., 100 nm), access to the absorptive surfaces of the small intestine that is vital to enhance diffusion into the non-polar (hydrophobic) environment of the brush-like projections of intestinal mucosal cells may be inhibited (Rigby & Schwarz, 2001). Hence, lipase enzymes secreted via the pancreatic duct of the pancreas binds to the surface of the emulsion oil droplets and breaks down triglycerides from the bulk of the lipid material to yield fatty acids and β-monoglycerides that can dissolve to some extent in the aqueous environment as a result of its charged groups and strong polar regions.

The products of the abovementioned process involve a long-chain hydrocarbon with non-polar regions, which spontaneously operate with the bile salts (of similar properties) to form mixed micelles. There is an essential small aggregate composition of free fatty acids, bile salts, and β-monoglycerides that have an interior with a non-polar region and an exterior with a polar (charged) region, within the aqueous phase (Meydani & Martin, 2001). The hydrophobic core of the mixed micelles dissolves the fat-soluble vitamin and other associated lipid substances that are non-polar. Mixed micelles are generally too small (4–10 nm in diameter) and can gain proximity to the border brush membrane of the intestinal mucosa, thus enhancing the diffusion of their contents into and across membranes (Combs, 2008).

Vitamins utilization: transport and cellular uptake

Like in vitamin absorption, the post-absorptive transport mechanism of vitamins also differs in accordance with their unique physical and chemical properties. Additionally, the dissolution of vitamins in a polar environment of blood plasma and lymph is another determinant factor for the transport of vitamins from its absorption site (small intestine) to the peripheral organs and liver (Albahrani & Greaves, 2016). As a result of the insolubility of fat-soluble vitamins in these transport environments, they rely solely on carriers that are soluble in these aqueous phases for their transport (Gil, Plaza-Diaz & Mesa, 2018). These vitamins, however, are coupled with blood lipid-rich chylomicrons (lipoprotein) that are largely synthesized in the intestinal mucosal cells, which serve to transport the absorbed fat-soluble vitamins to other lipoprotein carriers such as very-low-density lipoproteins (VLDL) and high-density lipoproteins (HDL) while some (vitamin A and D) are transported by specific binding proteins (retinol-binding proteins (RBP), transcalciferin, cellular retinoic acid-binding protein (CRABP), cellular retinol-binding protein (CRALBP), and vitamin D receptor) from the liver to peripheral tissues (Feingold, 2000; Combs, 2008; Zhang, Zhao & Huang, 2012; Bouillon et al., 2020). Lipoproteins are involved in the lipid-soluble vitamin transport given its molecular characteristics of having a lipid-protein combination, with the solubility properties of protein that exhibits dissolution in aqueous environment of the blood plasma (Saba & Oridupa, 2012; Axmann et al., 2019; Christie, 2021).

However, for the water-soluble vitamins, while several absorbed vitamins (niacin, vitamin C, riboflavin in the form of flavin mononucleotide, biotin, folate, and pantothenic acid) are freely transported in plasma solution via the erythrocyte membranes, some water-soluble vitamins (free riboflavin and pyridoxine) are transported by weak, non-specific binding with albumin and immunoglobulins (for free riboflavin only) which may be displaced by other compounds that can also bind to the protein. Others (vitamin B12 and riboflavin) are carried by specific binding proteins such as flavoproteins, riboflavin binding protein, and transcobalamin I, II, and III (Combs, 2008; Gil, Plaza-Diaz & Mesa, 2018). The absorption mechanism and post-absorptive transport of vitamins in the body are shown in Table 4.

Table 4 Absorption mechanism and post-absorptive transport of vitamins in the body.

Vitamin	Enteric vitamin absorption	Vehicle	
	Enterocytic metabolism	Site	Absorption	Transport	Cellular uptake	
A	Micelle-dependent diffusion	D, J	Retinol: Passive diffusion Carotenoid: Facilitated diffusion	Lipoprotein carrier (chylomicron), specific binding protein (RBP, CRBP. CRBPII, IRBP, CRALBP & CRABP)	Facilitated transport	
D	Micelle-dependent diffusion	D, J	Passive diffusion	Lipoprotein carrier (chylomicron), Specific binding protein (Transcalciferin, vitamin D receptor)	Passive diffusion	
E	Micelle-dependent diffusion	D, J	Passive diffusion	Lipoprotein carrier (Chylomicron, VLDL/HDL), specific binding protein (vitamin E BP), Erythrocyte carried (Erythrocyte membranes)	Facilitated transport	
K	K2 and K3: Micelle-dependent diffusion	D, J	Passive diffusion	Lipoprotein carrier (Chylomicron, VLDL/HDL)	Passive diffusion	
K1: Active transport	D, J	
C	Active transport	I	Passive diffusion, Carrier-mediated diffusion	Free in plasma (Erythrocytes)	Simple diffusion	
Simple diffusion	D, J, I	
B1	Active transport	D	Passive diffusion	Free in plasma (Erythrocytes)	Passive diffusion, Carrier-mediated transport	
Simple diffusion	J	
B2	Active transport	J	Carrier-mediated diffusion, Passive diffusion	Free in plasma (Erythrocytes), non-specific carrier proteins (Albumin, Immunoglobulins), Specific binding proteins (Riboflavin BP, flavoproteins)	Carrier-mediated transport	
B3	Facilitated diffusion	J	Passive diffusion, Facilitated diffusion	Free in plasma (Erythrocytes)	Passive diffusion	
Simple diffusion	J	
B5	Facilitated diffusion	J	Passive diffusion	Free in plasma (Erythrocytes)	Carrier-mediated transport	
Simple diffusion	J	
B6	Simple diffusion	J	Passive diffusion, Carrier-mediated diffusion	Erythrocyte carried (Erythrocyte membrane), non-specific carrier proteins (Albumin)	Passive diffusion	
B7	Facilitated diffusion	J	Facilitated diffusion, Passive diffusion	Free in plasma (Erythrocytes)	Facilitated transport, Carrier-mediated transport	
Simple diffusion	D, J	
B9	Simple diffusion	J	Passive diffusion, Carrier-mediated diffusion	Free in plasma (Erythrocytes)	Facilitated transport	
B12	Active transport	I	Passive diffusion	Specific binding proteins (Transcobalamin I, Transcobalamin II, Transcobalamin III)	Passive diffusion	
Simple diffusion	D, J	
Note:

Abbreviations: D, duodenum; J, jejunum; I, ileum; VLDL, Very low density lipoprotein; HDL, High density lipoprotein; BP, Binding protein; RBP, Retinol BP; CRBP, Cellular RBP; CRBPII, Cellular RBP, type II; IRBP, Interstitial RBP; CRALBP, Cellular retinal BP; CRABP, Cellular retinoic acid BP (Source: Adapted from Combs, 2008).

The extent and actual rate of cellular uptake of food-sourced vitamins vary due to some factors already mentioned above (differing losses, physiological effects, varying biopotencies, dietary effects, and health status) (Combs, 2008; Uribe, Garcia-Galbis & Espinosa, 2017). As a result, the human body would not entirely utilize all the vitamins in foods. The quantification of the total amount of food-sourced vitamins (obtained by analysis), therefore, would not equate and/or guarantee the actual nutritional value of those vitamins, because of these absorption impediments at cellular level (Zuvarox & Belletieri, 2021; Ruiz, 2021). In terms of vitamin retention and distribution, generally, their physical and chemical properties also determine the site of storage or distribution. Overall, the fat-soluble vitamins are significantly retained in the body and tend to be stored in lipid-rich tissues like the liver and adipose. Through enterohepatic circulation, fat-soluble vitamins would be excreted with the faeces, except for some water-soluble metabolites of vitamin A (retinoic acid), vitamin E (Simon metabolites, α-tocopheronic acid and its lactone), and vitamin K intermediate (menadione), also excreted in the urine despite being retained in lipophilic environment (Wallert et al., 2014; Shearer & Newman, 2014). In contrast, the water-soluble vitamins appear not to be retained in significant amounts in the body and are rapidly excreted through urine. Interestingly, an exception applies to vitamin B12 which can accumulate in the liver in significant quantities under normal circumstances (Gil, Plaza-Diaz & Mesa, 2018).

Characteristic interactions associated with food-sourced vitamins

Nowadays, increased nutritional awareness has significantly influenced the approach to food production and consumption towards a personalized optimal health experience rather than its original archaic perspective—food for survival. Specifically, food is a complex system with multitudes of variable chemical components (compounds) coexisting together (Sikorski & Haard, 2007). Foods with such complex systems are bound to have ubiquitous molecular interaction between macronutrients, micronutrients and even ions, exhibiting different interactivities such as hydrophobic interaction, covalent bonding, hydrogen bonding (bridges), electrostatic or ionic interaction, π–π stacking, and coordination force, among others (Gao et al., 2016). Food-sourced vitamins are diet-oriented and of plant and animal origin. Besides having less potential for toxicity, food-sourced vitamins are important because of their distinct advantage over vitamins in dietary supplement. The former is naturally associated with other food substances of beneficial effect that may enhance their absorption and utilization at the cellular level (Weininger, 2021). Though dietary supplement cannot replace a healthy diet, there seems to be a conflicting compromise between food-sourced vitamins, as some components present in a food matrix can enhance vitamin absorption, others may hinder its utilization in the body even up to cellular level. These phenomena in the food matrix have arisen due to interactions between vitamin-vitamin, as well as vitamin–mineral interactions. In other words, the molecular interaction among food components forms the basis for differences in the stability and functional characteristics of food-sourced vitamins.

(a) Vitamin–vitamin interaction

Multiple studies have shown that a combination of some B-complex (thiamine, riboflavin, folic acid, and cyanocobalamin) and vitamin C can produce detrimental effects. Most of these interactions usually occur in beverages fortified with these vitamins or when vitamins are used to restore food products presented in a liquid state such as fruit juices or soft drinks (Paquin, 2009). The mutual interaction amongst these five vitamins has been established, as the principal ones are briefly discussed below.Vitamin B12–folic acid:

The metabolic interaction between the two vitamins occurs during a deficiency of either vitamin in the catalytic transformation of homocysteine to methionine by the methionine synthase enzyme, which results in anaemia (Shane, 2008; Kumar et al., 2017). Studies have shown that there is a reduction in the activity of the vitamin B12-dependent methionine synthase enzyme with a proportionate deficiency in vitamin B12. This invariably results in a deficiency of functional folate because a higher concentration of folate has been confiscated or trapped as 5-methyl-tetrahydropteroyl-poly-glutamate (Mahmood, 2014). Additionally, the combined effect of vitamin 12 and folic acid deficiencies in fish (Labeo rohita) was found to increase the development of anaemia, making it more pronounced.Vitamin C–folic acid:

The metabolic interaction between the two vitamins initiates bond breakage of folic acid in a solution containing vitamin C and folic acid, as a result of the reducing property of vitamin C. The cleavage of folic acid has been demonstrated to be slowest in a very low acidic medium (pH 6.56–6.70) and most rapid in a high acidic medium (pH 3.0–3.3) (Ottaway, 1993; Schnellbaecher et al., 2019; Wusigale & Liang, 2020).Vitamin B12–Vitamin C:

The stability of vitamin B12 vitamers (hydroxocobalamin, methylcobalamin, and cyanocobalamin) is affected by interaction with vitamin C (ascorbic acid) in an aqueous solution. The instability is owed to the degradation losses of these vitamers in the presence of ascorbic acid which appears to be pH-dependent, as higher losses are recorded at low acid to neutral pH while lower losses are obtained in a very low acid range (Ottaway, 1993). According to Lie, Chandra-Hioe & Arcot (2020), methylcobalamin exhibited the greatest degradation losses (70–76%) in the presence of ascorbic acid, followed by hydroxocobalamin and cyanocobalamin. However, the degradation of vitamin B12 is maximum at pH 5.0 which is an indication of peak interaction between vitamin B12 and ascorbic acid. The increased interaction (degradation) of vitamin B12 at pH ≤ 5.0 is a result of the gradual ionization of ascorbic acid to form the ascorbate monoanion while the reduced interaction at pH ≥ 5.0 is owed to the loss of ascorbate monoanion by oxidation (Ahmad et al., 2014).Vitamin B12–Thiamine:

Studies have shown that the interaction between vitamin B12 and thiamine are mutually dependent. Vitamin B12 vitamers have been demonstrated to be unstable in the presence of thiamine by exhibiting degradation losses of about 48% for methylcobalamin, 24% for hydroxocobalamin, and 6% for cyanocobalamin (Lie, Chandra-Hioe & Arcot, 2020). The degradation of vitamin B12 is attributed to the formation of 4-methyl-5-(β-hydroxyethyl) thiazole during thiamine decomposition. In other words, the rate of vitamin B12 breakdown increases with the degradation of thiamine (Ottaway, 1993).Riboflavin-vitamin C:

Riboflavin has been found to act as a light-energy receptor, catalyzing the oxidation of ascorbic acid when exposed to light. This reaction suggests the increased rate of vitamin C degradation (loss) in food products like milk in the presence of sunlight (Ottaway, 1993; de La Rochette et al., 2000). This implies that photo-oxidation of vitamin C is significantly reduced in the absence of riboflavin (Yoshimoto et al., 2020).Riboflavin-folic acid:

The interactive effect of riboflavin and folic acid is significantly influenced by pH and light, which results in the degradation of folic acid by oxidation reaction (Ottaway, 1993). In other words, the extent of photo-degradation of folic acid in the presence of riboflavin is pH-dependent, since riboflavin’s stability is also pH-dependent (Akhtar, Khan & Ahmad, 2000). The interaction between riboflavin and folic acid appears to be enhanced as pH approaches neutrality from the acidic region since maximum degradation was recorded between pH 6.2–6.5. Moreover, as pH increases towards the alkaline region, the interaction of riboflavin with folic acid is greatly suppressed, because of nucleophilic attack on riboflavin, thereby resulting in a decrease in the rate of degradation of folic acid (Akhtar, Khan & Ahmad, 2000).Thiamine-Folic acid:

The mutual interaction between these vitamins has been shown to have a great influence on folic acid stability. The degradation products of thiamine, especially hydrogen sulfide can accelerate the decomposition rate of folic acid in solutions (Schnellbaecher et al., 2019). This reaction occurs particularly between the low acidic to neutral pH range (pH 5.7–7.0) (Ottaway, 1993). Some studies have also shown that the presence of UV light significantly promotes folic acid degradation (Schnellbaecher et al., 2019).Riboflavin–Thiamine:

The mutual interaction between riboflavin and thiamine has been found to result in the oxidation of thiamine by riboflavin, leading to the formation and precipitation of thiochrome in an alkaline environment. However, this reaction is predominantly found in solutions containing only the B-vitamins (Ottaway, 1993). On the other hand, in solutions containing thiamine, riboflavin and ascorbic acid, an alteration in riboflavin is seen by the reaction of thiamine (in the form of thiamine hydrochloride) in high concentration (relative to riboflavin) results in the oxidative action of thiamine by riboflavin, to form and precipitate chloroflavin as the degradation product of riboflavin (Schnellbaecher et al., 2019).

(b) Vitamin–mineral interaction

The interaction of vitamins and minerals has been reported in different metabolic situations, with emphasis on the nutritional importance these interrelationships provide, specifically in the maintenance of vital physiological role and biochemical processes that are crucial for human living. This mutual interaction can perform its biochemical-physiological function in various ways such as the action of minerals on vitamin metabolism, the action of vitamin on mineral metabolism, and/or the action of both micronutrients in cellular protection (Vannucchi, 1991).

Briefly, zinc has been found to significantly influence and participate in numerous aspects of vitamin A metabolism such as in its absorption, transportation, and utilization (Christian & West, 1998). However, it is well-known that vitamin D enhances intestinal absorption of phosphate and calcium, stimulates mobilization of bone calcium, increases the blood concentration of calcium, as well as enhances renal reabsorption of calcium in the distal tubule (DeLuca, 1986; Bowen, 2007; Ernst, 2016; Bowen, 2019). Further, vitamin E and selenium function together to protect biological membranes against lipid peroxidation. Both micronutrients have been reported to participate in the prevention of nutritional muscle degeneration known as muscular dystrophy (Ottaway, 1993). Vitamin C interacts with iron to enhance adequate absorption of iron from diets. The interaction gives rise to the formation of an iron chelate complex at acid pH, which enhances iron solubility in the small intestine, thereby increasing its utilization in the duodenum (Lynch & Cook, 1980; Beck, 2014; Li et al., 2020). On the other hand, the synergistic action of vitamins and minerals in the protection of organism at the cellular level has been recorded for vitamin E, C, carotenoid and selenium, as well as other important trace elements as an antioxidant component in diets. Specifically, these indispensable antioxidant vitamins and selenium, interact to protect tissues from free-radical degradation by quenching active (singlet) oxygen species, sequestering elements by chelation, and scavenging reactive oxygen species (DeLaval, 2007; McDowell et al., 2007; Ofoedu et al., 2021).

(c) Other interactions

Besides vitamin–vitamin and vitamin–mineral interactions, vitamins can also interact with series of other ingredients or components in food, which may affect its stability over a period. The rate of vitamin deterioration either by complete or partial loss is influenced by (i) Temperature (ii) Moisture (iii) Oxygen (iv) Light (v) pH (vi) Oxidising and reducing agents (vii) Other food components such as Sulphur dioxide (viii) Drug (for vitamin supplements) (ix) Combinations of the above (Godoy, Amaya-Farfan & Rodriguez-Amaya, 2021). Amongst these factors, temperature, moisture, oxygen, pH, and light appears to be the most important factors that affect vitamin stability in food.

Furthermore, as a heterogeneous group of a chemical compound, vitamin stability varies from being relatively stable, for example, in vitamin B3, to relatively unstable, in vitamin B12 which is unstable in alkaline and acid solutions, as well as degraded by light (highly photosensitive) and oxidizing or reducing agents. However, vitamin degradation can also occur during processing, specifically to foods subjected to heat treatment, and naturally during storage, for example, during the gradual loss of vitamin C content in packaged foods stored over time. As a result of this, food products are subjected to vitamin enrichment or fortification (Ottaway, 1993). The interactions between food-sourced vitamins with other components (both intrinsic and extrinsic factors) are briefly illustrated in Fig. 4. It shows how some factors such as heat (temperature), light, minerals, vitamins, pH, oxygen, and water can interact with food-sourced vitamins.

Figure 4 Factors that interact with food-sourced vitamins.

Vitamin deficiencies

Vitamin deficiency also known as avitaminosis is a well-known nutritional-oriented situation that occurs when the human body gets deprived of certain amounts (of vitamins) relative to its needed requirement (Sinha, 2003; Blanco & Blanco, 2017). Essentially, if the human body were to maintain an adequate vitamin status, there are a variety of vitamins that have to be supplied in different amounts. This has to come from a balanced diet, which is necessary since no single food source possesses all the required vitamins (FAO/WHO, 2019). Table 5 shows that vitamin deficiency can cause a variety of health-related problems. Further, vitamin deficiency is categorized in two ways, namely: primary and secondary deficiencies (Combs, 2008). Specifically, the primary deficiency refers to when the reduction in vitamin status occurs as a result of failures to ingest adequate quantities of the vitamin necessary to meet physiological needs. On the other hand, secondary deficiency refers to when a reduction in vitamin status is owed to failure to absorb or utilize a vitamin post-absorptively, possibly due to age or health status (Johnson, 2020).

Table 5 Vitamin deficiency.

Vitamin type	Deficiency	
A	Bitot spot and Nyctalopia (Night blindness), Xerophtalmia, dry mucous memberanes, keratinization of the corneal epithelium	
D	Osteoporosis, Osteomalacia in adults, and rickets in children	
E	Retrolental fibroplasia (Retinal degeneration), Vision problems, weakened immune system, nerve & muscle damage.	
K	Bleeding disorders	
C	Scurvy	
B1	Beriberi and Wernicke-Korsakoff syndrome	
B2	Ariboflavinosis resulting in sore throat, skin disorders, edema of the mouth & throat	
B3	Pellagra	
B5	Deficiency is rare but may include symptoms like fatigue, insomnia, depression, vomiting, stomach pains, upper respiratory infections, etc.	
B6	Deficiency is rare but may include symptoms like skin rashes, cracked lips, a glossy tongue, tiredness, mouth sores, nerve pain, confusion, etc.	
B7	Deficiency is rare but may include symptoms like hair loss, scaly red rash on the face, etc.	
B9	Spina bifida and anaemia	
B12	Neurological difficulties, macrocytic anaemia, and neural-tube defects	
Note:

Source: Adapted from Combs (2008); Johnson (2020).

The potential causes of vitamin deficiency in humans would vary across individuals, cultural groups, and beliefs. In some cases, also, it can be linked to psychosocial and technological reasons such as poor food habits, poverty, ignorance, lifestyle as well as lack of a balanced diet (Singh & Singh, 2008). Additionally, the lack of vitamin-rich foods (consumption of highly refined foods), vitamin destruction (during storage, processing, and cooking), food taboos and fads, apathy (lack of incentive to prepare adequate meals), and anorexia (dental problems, homebound elderly, infirm) can also be listed to contribute towards vitamin-associated challenges across individuals (Combs, 2008; Carr & Rowe, 2020). Besides, deficiencies linked to biological causes can also include poor digestion (absence of stomach acid), malabsorption, increased metabolic need, impaired metabolic utilization, and increased vitamin excretion (Tsiaras & Weinstock, 2011; Griffiths, 2013).

Vitamin toxicities

Compared to vitamin deficiency, the situation of vitamin toxicity is rare. This is because it is well understood that the human body can remove excess vitamins, like water-soluble vitamins, which are not stored. However, vitamin toxicity is still a problem, and of great concern, especially in the like of megavitamin treatment (Evans & Lacey, 1986; Sanders, 2002). This was what Davidson (1984) referred to as an offshoot of orthomolecular medicine wherein symptoms are treated with massive doses of vitamins. Although megavitamin treatment would be justifiable in reality only for a few conditions, vitamins in large doses via self-medication would result in severe toxic complications. There is ample evidence of serious toxic effects of megavitamin treatment, for example, in the United States where a substantial number of adults have treated themselves (Davidson, 1984).

Table 6 reveals vitamin toxicities and we can see there are serious human health implications. Briefly, situations of toxicities can arise in both fat- and water-soluble vitamins (Evans & Lacey, 1986; Buehler, 2011; Delaney & Barke, 2017; Picincu, 2021). For instance, vitamin B7 (biotin) would have no direct toxicity, although not impossible if too many biotin supplements are taken (Carling & Turner, 2018; Anonymous, 2021a). However, high circulating levels of biotin can interfere with many commercially available immunoassays that utilize the interaction involving streptavidin (Anonymous, 2021b). It should be noted that when taking a vitamin B complex, more caution is required with vitamins B5 and B6 considering that the former may cause vascular dilation, itching, nausea, headache, and allergies (Anonymous, 2021b). However, some reports have shown the toxicity of Vitamin B6 and vitamin C to respectively cause nerve impairment and gastrointestinal upset, as well as diarrhoea. On the other hand, there is a high chance of fat-soluble vitamins being toxic to the body especially with an intake of supplements since fat-soluble vitamins are stored in the liver (Delaney & Barke, 2017).

Table 6 Vitamin toxicity.

Vitamin type	Toxicity	
A	Overdose can lead to Raised intracranial pressure (“Pseudotumour cerebri”), Chronic liver disease, Hair loss, Ingrowing toe nails resistant to treatment; Tenderness of bones; Birth defects; Weight loss; Anorexia; Alopecia; Diplopia	
D	Overdose can lead to Hypercalcaemia, Hypertension, Hypercalciuria, Renal calcinosis, Metastatic calcification, cardiovascular damage	
E	Overdose can lead to increased anticoagulant action of warfarin, gastrointestinal distress, gaemorrhagic toxicity, nausea, muscular pain, fatigue, double vision, creatinurea	
K	Overdose can lead to Haemolytic anaemia and Neonatal jaundice	
C	Overdose can lead to oxalate stones in predisposed individuals, possible teratogenesis and carcinogenesis, a multiplicity of minor idiosyncratic symptoms	
B1	Overdose can lead to stomach upset, nausea, vomiting, and diarrhea	
B2	No known toxicity since higher intakes are excreted in the urine and not stored	
B3	Overdose can lead to peptic ulcer, Alopecia, Arrhythmias, Hepatotoxicity, Hypotension, and Pruritus	
B5	Common side effects when administered can include abdominal pain, constipation, creatine phosphokinase (CPK) increase, dizziness/weakness, flulike illness, joint/muscle pain, nausea, pancreatitis, sore throat	
B6	Overdosage can lead to dependency, peripheral sensory neuropathy, and ataxia, decrease in the therapeutic effect of levodopa	
B7	Overdosage can lead to skin allergies characterized by flushing, rashes, and itchiness; increased risk for anaphylaxis; insomnia, lower vitamin C and B6 levels, and excessive thirst and/or urination; eosinophilic pleuropericardial effusion	
B9	Overdosage can cause gastrointestinal defects, loss of appetite, mental confusion, nausea, seizures, skin reactions, and sleep disturbances	
B12	No known toxicity however vitamin B complex requires caution when taken with vitamins B5 and B6. The former (that is vitamin B5) may cause vascular dilation, itching, nausea, headache, liver damage, and inflammation of the gastric mucosa. The latter (that is vitamin B6) from 2 g or more can lead to movement and nervous disorders	
Note:

Source: Adapted from Delaney & Barke (2017); Davidson (1984); Evans & Lacey (1986); Picincu (2021).

Studies have shown that high levels of vitamin A known as hypervitaminosis A causes dry itchy skin, loss of appetite, swelling of the brain, and joint pain as well as liver damage, and coma in severe cases (Delaney & Barke, 2017). Of very important concern has been the vitamin D, where its excess in serum plasma has been defined as a level greater than 150 to 250 ng/ml, and where its intoxication would involve levels greater than 375 ng/ml. This brings the situation of Hypervitaminosis D, which is presented with such health implications as abdominal pain, cardiac arrhythmias, hypercalcemia, and hypovolemia from renal sodium and water losses, neuropsychiatric disturbances, as well as suppressed parathyroid hormone level (Charen & Harbord, 2020). In particular, hypercalcemia is considered among the key situations arising from vitamin D toxicity. Other early symptoms of vitamin D toxicity include gastrointestinal disorders like anorexia, diarrhoea, constipation, nausea (Alshahrani & Aljohani, 2013). In addition, there are three major hypotheses believed to be associated with vitamin D toxicity, which includes: (1) Raised plasma 1,25[OH]D concentrations result in increased intracellular 1,24[OH]D concentrations; (2) Vitamin D intake raises plasma 25[OH]D levels to concentrations, which exceed the DBP binding capacity, and free 25[OH]D directly affecting the gene expression once it enters target cells; (3) Vitamin D intake that raises the concentrations of many vitamin D metabolites (Alshahrani & Aljohani, 2013).

Can vitamin requirements be met by food-based approaches?

It is well-known that the available foods in a given region are determined by the agricultural practices, ecological, climatic, socio-economic and cultural factors that influence the dietary pattern of the people (Caswell & Yaktine, 2013; Singh & Singh, 2017; Tandzi & Mutengwa, 2020). In the developed nations, almost all the nutritional needs of their population groups are met or even exceeded by dietary patterns. However, that is not the case in the developing or under-developed nations where food choices are restricted due to limited food production capacity, poor purchasing power, and/or abnormal cultural practices such as food taboos, fads, and fallacies (FAO, 2001; McNamara & Wood, 2019). In view of this, nutritional needs are met when people have access to a variety of nutritious foods, and in sufficient amount.

Importantly, through a combination of variety of foods in different proportions, a healthy diet can be achieved. In order to attain the nutritional requirement, ascertaining the range of specific food intake in a particular combination would be difficult, given the population density, economic constraint, and level of food production in a certain ecology which may restrict appropriate nutritional adequacy (WHO/FAO, 2003; Leitzmann, in press). Therefore, the formation of a food-based approach that does not only consider energy and protein adequacy but also an avenue that can meet the nutrient requirement and micronutrient density of the diet through a combination of various foods is very fitting (de Ridder et al., 2017; FAO/WHO, 2019). This can be achieved by discovering diet-related public health problems (malnutrition or certain deficiencies) in a locality or community and developing appropriate dietary guidelines.

Further, diet-related public health problem or micronutrient (vitamin) deficiency in most population groups especially in the developing countries is largely due to subsistence consumption of cereal grain or root and tuber-based diets as staples, which provide energy and protein with inadequate amino acid balance, but very limited essential micronutrients (FAO, 2001). The inclusion of a high micronutrient density diet such as pulses, fruits and vegetables (including green leafy vegetables) into staple foods is the most fitting way of combating micronutrient/vitamin deficiency, and ensuring optimal nutrition for the population group (WHO/FAO, 1996; Tapsell et al., 2016). In addition, to meet adequate vitamin requirement of the body, different combination of foods from both plant and animal origin must be ingested to attain a healthy diet lifestyle. Studies have shown that foods of animal origin contain about 92% of the entire vitamins while that of plant-origin contain about 77% of total vitamin. Since no single food source contains all the vitamins, a combination of different foods (especially from plant and animal sources) in adequate quantities of water and oil, are vital to meet the body’s vitamin requirement.

Food-sourced vitamins: COVID-19 and era of micronutrient deficiency

Nutrition is an integral aspect of good health and development that is related to stronger immune systems, improved infant, safer pregnancy and childbirth, child and maternal health, lower risk of non-communicable diseases (such as cardiovascular disease and diabetes), and longevity (UNICEF, 2019). Today, the burden of malnutrition (such as micronutrient deficiency, undernutrition, and overweight) especially in developing nations and the deadly COVID-19 pandemic is ravaging most part of the world. Given this, the food system has been greatly explored to assess the potential role of bioactive compounds such as vitamins, in the fight against malnutrition and COVID-19 (Jovic et al., 2020; Galanakis, 2020). In addition, various attempts through nutrigenomics have been made with an emphasis on how different foods and their components can interact with specific genes to yield health benefits such as the reduced risk of neurodegenerative/metabolic/autoimmune diseases, and some cancers (Sikalidis, 2018). This can be achieved by developing a personalized diet as a therapeutic approach to delaying and preventing the onset of disease and ensure optimal maintenance of human health (Ronteltap et al., 2013; Neeha & Kinth, 2013).

Besides its role against micronutrient deficiency (malnutrition), the consumption of foods rich in vitamins has also been shown to boost the immune system to help fight off viruses (Naik, Thakare & Joshi, 2010; Gibson et al., 2012). This provided insights for several ongoing studies on the use of food bioactive ingredients for the support of the human immune system against infectious disease like COVID-19 (Galanakis, 2020). Specifically to viral infectious diseases, Jovic et al. (2020) highlighted the potential role of vitamins A and E in the fight against COVID-19 through their antioxidant effect, enhancing natural barriers, immunomodulation, and local paracrine signalling. Studies have also shown the beneficial role of vitamins C and D either through food or dietary supplements in the fight against COVID-19 (Dehghani-Samani, Kamali & Hoseinzadeh-Chahkandak, 2020). For instance, it is well recognized that vitamin C is an essential micronutrient needed for the development and repair of all body tissues, play a protective role given its antioxidant effect, supports the immune function (Carr & Maggini, 2017), and also limits the responsiveness of lower respiratory tract to infection, under certain conditions (Hemilä, 1997; Galanakis, 2020).

Dietary supplements containing these aforementioned vitamins may also be key to support the human immune system against COVID-19 (Shakoor et al., 2021). In this new era of a pandemic, it is predicted that consumers will greatly go for immune-boosting products soon. This corroborates the findings of a survey that almost one in five consumers mentioned immune system support as the main reason for buying a healthy product (Avis, 2021).

Food-sourced vitamins and dietary supplements: striking a balance?

Prior to the pandemic era, the use of dietary supplement has significantly increased over the decades especially in developed nations, contributing to the nutrient intake in the population (Lentjes, 2019). Originally, the ultimate goal of dietary/food/nutritional supplement is to act as a vehicle for delivering nutrients that may not be consumed in adequate amounts. Dietary supplements are usually micronutrients in the form of vitamins, minerals, and/or their combinations with other substances prepared in the form of capsules, liquid, pills, tablets, etc. and administered during nutrient deficiency situations (Mensink et al., 2013).

To reiterate, urbanization and some sort of modern dietary pattern aimed at boosting the immune system and staying healthy could be the driving force behind the increased use of dietary supplements (de Jong et al., 2003; Lentjes, 2019). Dietary supplements are known to contain isolated nutrients in high concentration which is capable of increasing the total nutrient intake by raising the concentrations of the respective nutrients specifically vitamins, in blood plasma or serum (Lentjes, 2019). This unhealthy lifestyle associated with nutritional excesses can cause series of lifestyle-induced diseases such as diabetes, stroke, heart disease, obesity, etc. (Minichi & Bland, 2013). For example, high total nutrient intake of retinol (≈2,500 µg/day) in combination with total nutrient intake of vitamin D (<11 µg/day) has been linked with fractures in post-menopausal women (Caire-Juvera et al., 2009).

Notably, dietary supplements cannot replace a balanced healthy diet because a diet containing, for instance, whole grains, sufficient protein, healthy fats, fruits and vegetables can be enough to meet the daily adequate vitamin intake needed for good health (FAO, 2001; Kearney, 2010; Shao et al., 2017). However, vitamins are only required in certain amounts (µg or mg) and therefore higher amounts of some vitamins through supplementation could contribute to increased nutrient intake, thereby causing harm (FAO/WHO, 2004; Molnar & Gair, 2015). Although dietary supplements are used to augment dietary gaps especially during nutritional/micronutrient deficiency, studies have shown that nutrients from food are most important (Shenkin, 2006; Anonymous, 2021c). Table 7 shows the usual amount of vitamin intake from various foods and the prevalence of potential dietary supplementation. The information presented in this Table 7, reveals how the major vitamins directly connect with food sources, specific to average amount (of vitamins), considering mean daily intake, recommended dietary allowance (RDA), upper limit (UL) per day as well as estimated amount of supplementation (EAS) (Drake, 2017; Delaney & Barke, 2017; Harvard Medical School, 2020).

Table 7 Vitamin intake from various foods and the prevalence of potential dietary supplementation.

Vitamin	Food source	Average amount of vitamin in food	Mean daily intake from food	Mean RDA	UL per day	EAS (%)	
A	Milk, whole	6.2 µg	424 µg RAE	800 µg	3,000 µg	47%	
Cheddar cheese	7.1 µg	
Chicken liver	308 µg	
Beef liver	679 µg	
D	Sardines, canned in oil	1.15 µg	2.0 µg	18 µg	50 µg	89%	
Egg, large	1.03 µg	
Beef liver	1.05	
Milk, non-fat, vitamin-D fortified	3.10 µg	
E	Sunflower seeds	7.4 mg	6.8 mg	15 µg	1,000 mg	55%	
Almonds	6.8 mg	
Peanuts	2.2 mg	
Sunflower oil	5.6 mg	
K	Broccoli	160 µg	86.7 µg	105 µg	Not known	17%	
Cabbage	34 µg	
Ground beef	6 µg	
Spinach	27 µg	
B1	Breakfast cereals, fortified	1.5 mg	0.9 mg	1.2 mg	Not known	25%	
Pork, broiled	0.4 mg	
White rice, enriched	1.4 mg	
Tuna, cooked	0.2 mg	
B2	Beef liver	2.9 mg	1.7 mg	1.2 mg	Not known	NSR	
Egg. Scrambled	0.2 mg	
Plain yoghurt, fat free	0.6 mg	
Instant oats, fortified	1.1 mg	
B3	Turkey	10 mg	18.5 mg	15 mg	35 mg	NSR	
Whole wheat bread	1.3 mg	
Cereal (fortified)	20 mg	
Tuna	8.6 mg	
B5	Avocado	1 mg	Not known	5 mg	Not known	–	
Fish, trout	1.9 mg	
Egg	0.7 mg	
Yoghurt, plain non-fat	1.6	
B6	Chickpeas	1.1 mg	1.6 mg	1.5 mg	100 mg	NSR	
Tuna, fresh	0.9 mg	
Potatoes	0.4 mg	
Spinach	0.1 mg	
B7	Banana	0.2 µg	Not known	30 µg	Not known	–	
Pork chop	3.8 µg	
Broccoli	0.4 µg	
Sweet potato	2.4 µg	
B9	Beef liver	215 µg DFE	213 µg DFE	400 µg	1,000 µg	48%	
Spinach	131 µg DFE	
Broccoli	45 µg DFE	
Asparagus	85 µg DFE	
B12	Salmon	4.8 µg	4.3 µg	2.4 µg	Not known	NSR	
Milk, low fat	1.2 µg	
Tuna, canned	2.5 µg	
Breakfast cereal, fortified	1.5 µg	
C	Orange juice	93 mg	74.4 mg	83 mg	2,000 mg	10%	
Tomato	17 mg	
Grape juice	70 mg	
Romine lettuce	28 mg	
Notes:

Source: Adapted from Drake (2017); Delaney & Barke (2017); Harvard Medical School (2020).

Abbreviations: RAE, retinol activity equivalents; DFE, dietary folate equivalents; RDA, recommended dietary allowance; UL, upper limit; EAS, estimated amount of supplementation; NSR, no supplementation required.

Food (fresh or processed) is composed of nutrients such as vitamins, minerals, polyphenols, etc. necessary for the promotion of good health (Skerrett & Willett, 2010). According to Chen et al. (2019), adequate nutrient intake from food sources significantly reduced all-cause of cardiovascular disease (CVD) mortality, but the lower risk of death disappeared when the same nutrients were taken in the form of dietary supplements. This corroborates the findings that supplement use has been associated with mortality (Mursu et al., 2011). Furthermore, the study also showed that a lower risk of death was associated with adequate nutrient intakes of vitamin K and magnesium, only when these nutrients were sourced from food, but the lower death risk was lost when they were consumed as supplements (Chen et al., 2019). In addition, lower risk of death from CVD associated with adequate intakes of vitamins A, K, and zinc were also reported, provided that nutrients were sourced from foods and the lower death risk was lost when nutrients were consumed as supplements (Chen et al., 2019; Upham, 2019). This corroborates the findings that nutrients are most potent when they are obtained from food (Magee, 2010; Anonymous, 2015; Chakrabarti, Guha & Majumder, 2018; Khoo et al., 2019). The findings of this research suggest that individuals and population groups without any nutritional deficiency may benefit when vitamins and other nutrients are obtained from a healthy balanced diet, of which same benefits may be lost if those nutrients are obtained from dietary supplements. This implies that dietary supplement cannot be a replacement for a healthy diet and should not be a frequent part of the modern lifestyle of personalized diet.

Concluding remarks and future prospects

We have, in this perspective review, revisited food-sourced vitamins, and shed more light on their classification, metabolic functions, absorption, transport, deficiencies, and toxicities. Adding its importance as micronutrients, vitamins have to be provided in the required amounts for vital physiological functions and biochemical processes crucial to human living. How the physical and chemical properties of vitamins would influence the absorption of vitamins, vitamin transport, cellular uptake, and excretion, were also enumerated in this paper. As no single food source contains all the vitamins, a balanced healthy diet consumed from a variety of foods of plant and animal origin would help meet what the body needs. Besides performing a physiological function as an antioxidant, co-enzyme, hormone, and electron donor/acceptor (cofactors), the mode of vitamin absorption (simple, passive, facilitated, and carrier-mediated diffusion), a vehicle for vitamin transport (free in plasma, erythrocyte carried, lipoproteins, specific binding, and non-specific binding proteins), and mode of cellular uptake equally varies and depends on their associated chemical properties.

All vitamin absorption takes place in the small intestine. As water-soluble vitamins diffuse across the brush border membrane of intestinal mucosa into the bloodstream to the target cells and tissues, the fat-soluble vitamins require the help of bile acids to diffuse through the intestinal mucosa to the lymphatic system after being packaged as mixed micelles and transported by carrier-dependent proteins to peripheral tissues and liver. Besides the failure of ingesting vitamin-rich foods, vitamin deficiency can arise from impaired intestinal vitamin absorption owed to intestinal disease, genetic disorders in transport, excessive alcohol consumption, and interaction with drugs. Vitamins can also interact with each other, other food components, and certain factors such as pH, temperature, and light, to exhibit different degrees of absorption efficiency as well as partial or complete degradation. Additionally, the frequent and unchecked intake of vitamin supplements can induce vitamin toxicity, especially from fat-soluble types.

For the ultimate goal of safeguarding consumers’ health, obtaining vitamins naturally from the diet should be paramount to the intake from a vitamin supplement. Using micronutrient supplements to solve an unhealthy dietary pattern is a parochial approach to solving an unhealthy lifestyle. Therefore as the use of dietary supplement continues to increase, monitoring their contribution to diet, health, and disease is very crucial. The right amounts of vitamins must be contained in our daily meals to address the problem of micronutrient deficiency caused by nutritional insecurity especially in developing nations, as well as to support and maintain the sustainability of the human immune system against infections and health crises.

This perspective review has provided a foundational insight that can help future studies to dig deeper into understanding the intricacies that underpin how fat- and water-soluble vitamins interact with macronutrients within the food component matrix, and its influence on the absorption efficiency, transport, and cellular uptake. The direction of future studies should focus on developing technologies and processes that will make staple foods and frequently consumed commercial food products (for example, instant noodles, pasta, and some confectionery goods) as carriers and delivery system for vitamins especially the most common global deficient vitamins. This can be achieved by making vitamins as bioactive agents in commercial food products, such that vitamins maintain their effectiveness in an active form when it reaches the active site after ingestion, without variation in their efficacy.

Additional Information and Declarations

Competing Interests

Author Contributions

Data Availability

Charles Odilichukwu R. Okpala is an Academic Editor for PeerJ.

Chigozie E. Ofoedu conceived and designed the experiments, prepared figures and/or tables, authored or reviewed drafts of the paper, and approved the final draft.

Jude O. Iwouno conceived and designed the experiments, prepared figures and/or tables, and approved the final draft.

Ebelechukwu O. Ofoedu performed the experiments, prepared figures and/or tables, and approved the final draft.

Chika C. Ogueke analyzed the data, authored or reviewed drafts of the paper, and approved the final draft.

Victory S. Igwe conceived and designed the experiments, prepared figures and/or tables, and approved the final draft.

Ijeoma M. Agunwah analyzed the data, authored or reviewed drafts of the paper, and approved the final draft.

Arinze F. Ofoedum analyzed the data, prepared figures and/or tables, and approved the final draft.

James S. Chacha conceived and designed the experiments, performed the experiments, authored or reviewed drafts of the paper, and approved the final draft.

Onyinye P. Muobike conceived and designed the experiments, prepared figures and/or tables, and approved the final draft.

Adedoyin O. Agunbiade conceived and designed the experiments, performed the experiments, authored or reviewed drafts of the paper, and approved the final draft.

Njideka E. Njoku analyzed the data, authored or reviewed drafts of the paper, and approved the final draft.

Angela A. Nwakaudu performed the experiments, authored or reviewed drafts of the paper, and approved the final draft.

Nkiru E. Odimegwu performed the experiments, authored or reviewed drafts of the paper, and approved the final draft.

Onyekachi E. Ndukauba performed the experiments, prepared figures and/or tables, and approved the final draft.

Chukwuka U. Ogbonna performed the experiments, authored or reviewed drafts of the paper, and approved the final draft.

Joncer Naibaho performed the experiments, prepared figures and/or tables, and approved the final draft.

Maciej Korus analyzed the data, prepared figures and/or tables, and approved the final draft.

Charles Odilichukwu R. Okpala analyzed the data, authored or reviewed drafts of the paper, and approved the final draft.

The following information was supplied regarding data availability:

This is a review article, so raw data are not available.

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
