# Peer review of "Revisiting food-sourced vitamins for consumer diet and health needs: a perspective review, from vitamin classification, metabolic functions, absorption, utilization, to balancing nutritional requirements"

_PeerJ, doi:10.7717/peerj.11940_

## Round 0.1 · original submission · Major Revisions

Please address the concerns of all reviewers and amend your manuscript accordingly.

Reviewer 1 ·

Basic reporting

no comment

Experimental design

The "Survey Methodology" part seems vague, which need more details for the readers to fully understand how authors performed the survey. For examples, in line 159-160, "we applied discretion to strike a balance between the year of publication and the relevance of the information." Yet, it is not clear what is the specific strategy they use. What's the threshold for year of publication? How they determine the relevance? The authors should clarify this with details in the text.

Based on my understanding, the main research items, according to line 151 are vitamin absorption, transportation of vitamin, and cellular uptake of the vitamin. Yet, in my following reading of the paper, I got very limited new knowledge in these areas. Majority of the content goes to the detailed introduction of different vitamins which can be found in many popular science articels on the internet. I would suggest the authors focused more on the research findings in the specific research items they mentioned in the method.

As for the presentation of the data, I think there are too many figures, I would suggest the authors put the structures of different vitamins into one figure.

Validity of the findings

In line 589-592, the author claims "This paper provides insight for future reviews/studies, which should help to further establish the intricacies underpinning how fat- and water-soluble vitamins interact within the food component matrix, as influenced by its absorption, transport, and cellular uptake." However, I don't see enough content to support their conclusion. In my opinion, the review provides very limited novel insights or raise any unresolved questions in the field. The authors should discuss more about the key questions that need to be addressed as well as gaps that need to be filled in the future.

Additional comments

Generally, the review provide very comprehensive introduction to the food-sourced vitamins in regarding to their classification, sources, functions,etc. However, the review seems unbalanced, which loses focus on its scientific domain. The authors should reshape their writings to make it more suitable for a scientific review rather than a popular science article.

·

Basic reporting

The review meets the criteria of reporting and within the scope of the PeerJ journal.

Experimental design

The review is self-contained and meets the criteria of the PeerJ Journal.

Validity of the findings

The question is well-defined with relevant discussion and meets the criteria of the PeerJ Journal.

Additional comments

This review by Chigozie E Ofoedu et al describes the classification, metabolic functions, absorption, transport, deficiencies, and toxicities of vitamins, an essential micronutrient of our body. The detailed and orderly presentation is impressive. The review is organized logically into critical subsections which are well developed and explanatory to the goals set out in the beginning of the article. I am recommending accepting this article with minor revision.

1) The authors could consider compiling figures from 2 to 14 (structure of vitamins) to a single figure.

2) The manuscript could use some editing, since some sentences are difficult to read.

a) Line number 487-488 of the manuscript should be written as

“These vitamins, however, are coupled with blood lipid-rich chylomicrons that are largely synthesized in intestinal mucosal cells”

b) Line number 512-513 of the manuscript should be written as

“Essentially, if the human body were to maintain an adequate vitamin status, there are a variety of vitamins that have to be supplied in different amounts”

c) The full form of ATP (adenosine triphosphate) has been mentioned in line number 281, but the word ATP has appeared before this in line number 243.

d) Use punctuation mark (.) in line number 383 after references [4,25].

e) Line number 278-279 of the manuscript should be written as
“Also, niacin aids in nervous system function and digestion by participating in oxidation-reduction metabolic reactions, protects against neurological degeneration, and has lipid-lowering ability [4,11,28]”.

3) The authors could elaborate more on these sentences of the manuscript:

a) In Line number 122-123, what is “bloodstream-physiological activities”?

“Further, the water-soluble vitamins diffuse through the intestinal walls into the
bloodstream-physiological activities.”

b) In line number 242-243, the sentence could be framed as
“In addition, as a key player in the synthesis of neurotransmitters, it is also required for RNA, DNA, and ATP synthesis [11].”

Reviewer 3 ·

Basic reporting

no comments

Experimental design

no comments

Validity of the findings

no comments

Annotated reviews are not available for download in order to protect the identity of reviewers who chose to remain anonymous.

---

## Round 0.2 · accepted · Accept

All critiques were addressed and the revised manuscript is acceptable now.

Reviewer 1 ·

Basic reporting

no comment

Experimental design

no comment

Validity of the findings

no comment

Additional comments

The authors made a great effort to reshape their writings and the revised manuscript improved by addressing reviewers' concerns. I have no further comments.

·

Basic reporting

The manuscript meets the basic criteria of reporting.

Experimental design

The manuscript meets all the criteria set out by PeerJ.

Validity of the findings

Please see the General Comments section for details.

Additional comments

My all comments have been addressed by the authors and the manuscript can be accepted for publication.

Reviewer 3 ·

Basic reporting

none

Experimental design

none

Validity of the findings

none

Additional comments

Overall the revised manuscript was well prepared. The authors provided detailed descriptions and additional sections, which are interesting and significant. Now the manuscript is ready to be accepted, while still need minor revision before publication.

1, noticed an equal contributor were added as a cofirst author, but the names, labeling, the order, and footnote in the first page were confusing. There should be some mistakes.

2, The keywords (Line69): could be modified by being replaced with more appropriate words, removing “animal sources; plant sources” at least.

3, The subtitle in each section is not well organized. For example, in “Absorption of Vitamins” part, there is no “ a) ”.

4, Be careful with word typing, capital or lower case, and consistence, e.g. COVID-19 vs Covid-19.

5, I don’t think the references were well adjusted according to the journal standard, in both text and ref list. For example, the number of authors, name, journal abbreviation.

6, The figure 2 quality is not good. Those structure figs, some of them, should be well prepared. Screenshot even contains watermarks, are not acceptable.